# Elevation and land use shape soil entomopathogenic fungal communities in the Uluguru mountains, Tanzania: Insights from metagenomic and culture-based approaches

**Abel Jonathan Mussa**[1*], **Joseph O. Ruboha**[1,2], **Sija A. Kabota**[1,3], **Martin J. Martin**[4], **Maulid W. Mwatawala**[1]

**1** Department of Crop Science and Horticulture, Sokoine University of Agriculture, Morogoro, Tanzania, **2** Department of Agricultural Sciences, Mizengo Pinda Campus College, Sokoine University of Agriculture, Katavi, Tanzania, **3** Research, Consultancy and Publication Unit, National Sugar Institute (NSI), Kidatu-Morogoro, Tanzania, **4** Institute of Pest Management (IPM), Sokoine University of Agriculture, Morogoro, Tanzania

* abel.mussa@sua.ac.tz

## Abstract

### Background

Soil-borne entomopathogenic fungi (EPFs) support ecological regulation of pests, yet their distribution across tropical mountain agroecosystems is poorly characterized. The study conducted between April and December 2024, evaluated diversity and distribution of soil EPF along the Uluguru Mountains slopes in Morogoro, Tanzania.

### Methods

Twenty-four soil samples were collected from cultivated and fallow soils at low (518 m), medium (1100 m), and high (1700 m) elevations on the Uluguru slopes (Morogoro, Tanzania). Amplicon sequencing of the ITS region profiled fungal communities, and selective isolation with ITS barcoding confirmed cultivable taxa. Diversity indices, Bray-Curtis dissimilarity, Principal Coordinate Analysis (PCoA), and PERMANOVA evaluated patterns across elevation and land use.

### Results

Fourteen EPF species in 12 genera were detected, dominated by Ophiocordycipitaceae (56.1%) and Clavicipitaceae (37.8%). *Purpureocillium lilacinum, Metarhizium anisopliae, Clonostachys rosea,* and *Pochonia chlamydosporia* were widespread. Cultivated soils at medium- and high elevations showed greater richness and diversity (1.37 and 1.57) than fallows (0.64 and 0.48) respectively, while high-altitude fallows were strongly dominated by *Metapochonia suchlasporia*. Community composition clustered by land use, with elevation as a secondary driver (PERMANOVA

**Data availability statement:** All raw ITS metagenomic sequencing data generated in this study have been deposited in the National Center for Biotechnology Information (NCBI) Sequence Read Archive (SRA) under BioProject accession PRJNA1425464, and are publicly available at: https://www.ncbi.nlm.nih.gov/sra/PRJNA1425464. The dataset includes BioSample accessions SAMN55416426-SAMN55416435. Normalized read counts and associated sample metadata, including GPS coordinates, elevation, and land-use type, are provided in the Supporting information file. All data are publicly available without restriction.

**Funding:** This work was supported by the Sokoine University of Agriculture (SUA) internal research fund (Grant No. SUA/2024/AGROBIO/EPF). The funders had no role in study design, data collection and analysis, decision to publish, or preparation of the manuscript.

**Competing interests:** The authors have declared that no competing interests exist.

p = 0.06). Selected *P. lilacinum* and *C. rosea* species caused 10–50% mortality of *Spodoptera frugiperda* larvae in preliminary laboratory assays.

## Conclusions

Elevation and land use jointly structure EPF communities in the Uluguru Mountains. Some taxa showed preliminary pathogenicity in laboratory assays, indicating potential for future evaluation as biological control agents in smallholder farming systems. Public deposition of sequencing reads will facilitate reuse and benchmarking.

## Introduction

Entomopathogenic fungi (EPFs) are a diverse group of micro-organisms that naturally infect and regulate populations of more than 700 insect pests under laboratory, semi-field and field conditions, making them valuable in biological control programs [1]. They mainly belong to the order *Hypocreales* under phylum *Ascomycota*, that includes important families such as *Clavicipitaceae*, *Cordycipitaceae*, and *Ophiocordycipitaceae* [2,3]. Within these families, several genera are widely recognized for their ecological and applied significance, including *Beauveria*, *Metarhizium*, *Isaria* (syn. Cordyceps), *Lecanicillium* and *Purpureocillium* [4,5]. Notable species include *Beauveria bassiana (Bals.-Criv.)*, *Metarhizium anisopliae* (Metschn.), *Isaria fumosorosea* (Wize), and *Purpureocillium lilacinum* (Thom), which infect a wide range of insect hosts. Some EPF species have been extensively studied and developed into commercial mycoinsecticides against invasive pests such as the fall armyworm, *Spodoptera frugiperda* (J.E. Smith), which is now widespread in Tanzania and other parts of Africa [6,7].

Crop yield losses due to insect pests have been estimated at about 20% worldwide per annum [8]. Conventional reliance on synthetic insecticides has been associated with environmental pollution, ecosystem disturbances, human health risks, and high production costs [9,10]. EPFs provide an alternative, eco-friendly strategy for pest suppression. They infect insects through enzymatic degradation of the cuticle and production of secondary metabolites [2,11,12]. In addition, EPFs can persist in soils across different life stages, regulating insect populations while contributing to improved soil health [13,14]. Some species, such as *P. lilacinum*, also exhibit endophytic capabilities that enhance plant growth and nutrient uptake, thereby contributing to biodiversity conservation [15–17].

Environmental gradients and anthropogenic factors such as elevation-driven microclimatic variations, soil properties, and land-use practices strongly influence diversity and distribution of EPFs worldwide [18–21]. Global studies show that elevational gradients generate differences in temperature, humidity, and solar radiation, which in turn affect fungal physiology, spore germination, and pathogenic activity. This leads to localized variation in EPF communities even within the same ecological region [18,21–25]. For instance, *B. bassiana* and *P. lilacinum* occur across a wide elevational range, whereas *Metarhizium* spp. typically dominate warmer

lowlands, and *I. farinosa* is more common in cooler highlands [26,27]. Studies conducted in China confirm these patterns [18,21,27,28]. Land-use intensification, especially through tillage and agrochemical inputs, alters soil structure and chemistry, reducing microbial diversity across habitats [26,29]. Comparable studies further indicate that *Metarhizium* spp. tend to dominate in cultivated soils, while *Beauveria* spp. are more common in semi-natural habitats [30–32].

Despite this progress, major knowledge gaps remain, particularly in Africa. Surveys confirm the presence of native EPFs in Tanzania [33], Ethiopia [6] and Nigeria [34].

However, most studies focus on a few taxa, mainly *M. anisopliae* and *B. bassiana*, with little attention to altitudinal gradients, seasonal dynamics, or long-term persistence. Few studies explicitly link biodiversity indices with pest suppression under field conditions, leaving the functional role of EPFs diversity poorly understood. While soil chemistry and microhabitat drivers have been examined in Asia and Europe [19,21,26,27,35,36], such analyses are scarce in East African systems. Genomic approaches also remain underutilized, restricting insights into cryptic diversity and adaptive traits. Moreover, the interactions between EPF diversity and agroecological practices such as intercropping, mulching, or organic amendments remain underexplored. Addressing these gaps is crucial for harnessing locally adapted EPFs as reliable components of agroecological pest management. In Tanzania, research on soil-dwelling EPFs remains limited, with most studies focusing on pathogenicity tests of imported commercial species and little attention given to native isolates from insect cadavers or soils [33,37]. Existing studies remain fragmented and rarely assess how ecological drivers such as elevation and land use shape community structure.

The Uluguru Mountains are part of the Eastern Arc Mountains, a globally recognized biodiversity hotspot for their exceptional species richness and high levels of species restricted to the area [38,39]. The area is characterized by steep slopes, deeply dissected valleys, and vegetation types ranging from lowland woodlands to montane and cloud forests, interspersed with diverse agricultural systems [40]. Despite the recognized importance of EPFs in pest management and the global significance of the Uluguru Mountains as a biodiversity hotspot, systematic knowledge of EPF diversity and spatial distribution in Tanzanian soils is lacking, particularly across elevational and land-use gradients. This gap limits the potential to exploit native fungal resources for pest control, thereby increasing reliance on a limited set of local species and expensive imported strains that are often inconsistently effective and inaccessible to smallholder farmers [10].

This study investigated the diversity and distribution of soil-borne EPFs across different elevation gradients (low, medium, and high) and land-use types (cultivated and fallow) along the Uluguru mountain slopes.

We hypothesized that both elevation and land use significantly influence the diversity and distribution of soil-borne EPFs in the Uluguru mountain slopes. Specifically, we expected greater richness in undisturbed fallow soils compared to cultivated soils, and shifts in community structure along the altitudinal gradient, with thermophilic taxa dominating lower elevations and psychrotolerant species more prevalent at higher elevations. Identifying fungal taxa naturally abundant in different soils can reduce dependence on imported strains, enhance the effectiveness of IPM programs, and contribute to sustainable crop production [10,12].

## Materials and methods

### Description of the study area

The study was conducted along the slopes of the Uluguru Mountains in Morogoro, Tanzania, at three sites representing an altitudinal gradient from low to upper elevation: Sokoine University of Agriculture (SUA, 518 m a.s.l.), Langali (1100 m a.s.l.), and Nyandira (1700 m a.s.l.). The sites are located between 6.45°-7.51° S and 37.36°-37.45° E in the Morogoro Region. Due to elevation differences, the sites exhibit marked temperature variation, ranging from warm conditions at SUA (25–30°C) to cooler conditions in Nyandira (10–18°C). Rainfall also increases with altitude, from about 1000 mm annually in the lowlands to over 2000 mm in the upper slopes [38].

Soil samples were collected from both cultivated and fallow lands at each site to capture variation in land-use practices. All laboratory analyses related to pathogen isolation and molecular characterization were carried out at the SUA Pathology and Molecular Laboratory.

## Soil sample collection

A total of twenty-four soil samples were collected between April and November 2024 from three different altitudinal sites (SUA, Langali, and Nyandira), with eight samples per site; four from cultivated areas and four from fallow land, each separated by a minimum distance of 20 m (Fig 1 and S3 Table). At each microsite, a single composite soil sample was collected using a soil auger at a depth of 0–20 cm from three randomly selected points. The soil was then packed approximately 200 g in labelled polythene bags and transported to the laboratory for further processing [32,41]. We selected fallow soils had been left uncultivated for at least 2 years, allowing soil fungal communities to re-establish and reflect post-cultivation diversity, whereas cultivated soils represent current agricultural conditions. Geographical location coordinates were recorded using GPS and were used to map the study area using QGIS software.

## Metagenomic DNA extraction, amplification, high-throughput sequencing and quality control

Total metagenomic DNA was extracted from approximately 10 g of soil using the ZymoBIOMICS™ DNA Miniprep Kit (50 preps; Inqaba Biotec East Africa Ltd.) following the manufacturer's protocol with minor modifications [24]. DNA quality was

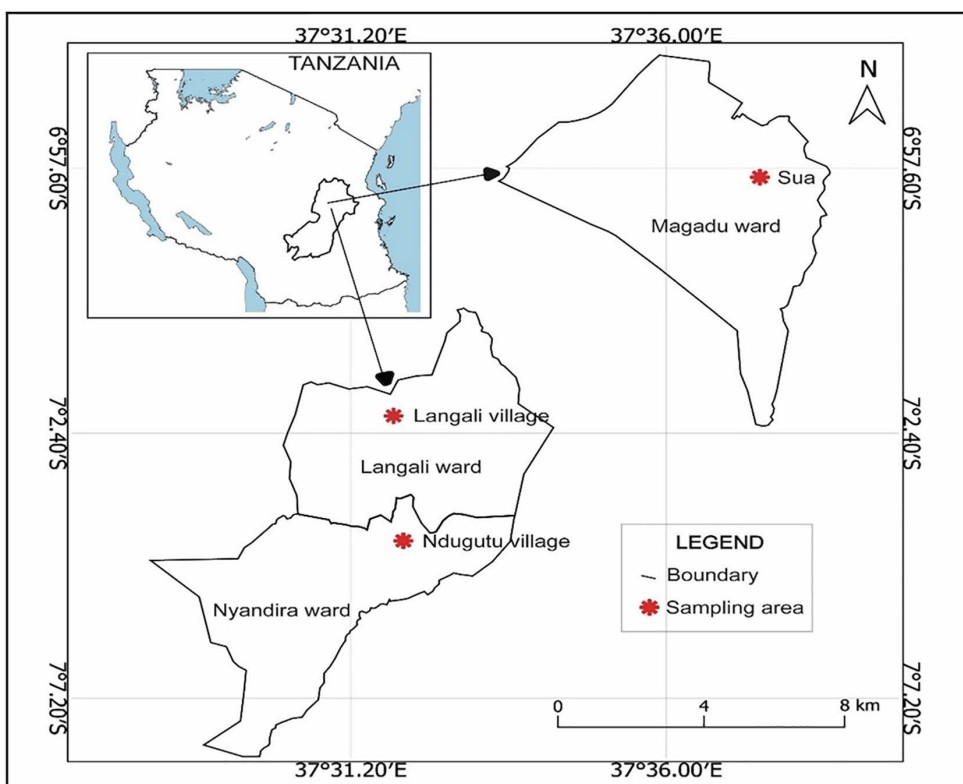

**Fig 1. Study area map showing the sampling stations.** The map was created by the authors using QGIS software (version 3.34.1). All spatial layers were derived from administrative boundary shape files obtained from the National Bureau of Statistics (NBS), Tanzania (Click here for NBS GIS data), a public national institution, and are publicly available for use under CC BY 4.0 or equivalent open license.

initially assessed using 1% agarose gel electrophoresis run at 80 V in TAE buffer (0.04 M Tris, 0.02 M acetic acid, 2 mM EDTA, pH 8.0), and concentrations were further measured using a Nanodrop 2000 spectrophotometer.

High-quality DNA samples were submitted to Macrogen Inc. (Gangnam-gu HQ, Seoul, Republic of Korea) for amplicon-based high-throughput sequencing (HTS). The internal transcribed spacer (ITS1-ITS2) region was targeted using the Illumina MiniSeq platform with a paired-end (PE) strategy, 301 bp read length, and an average sequencing depth of approximately 50,000 PE reads per sample [24].

Quality control of the sequencing reads was performed before pre-processing using FastQC v0.11.9. Poor quality bases and sequencing adapters were trimmed using Trim Galore version 0.6.4_dev powered by Cutadapt version 2.8 with Python 3.8.10 [42]. Quality Phred score cutoff of 20 was used with the following additional settings: quality encoding type selected: ASCII+33, adapter sequence: 'CTGTCTCTTATA' (Nextera Transposase sequence; auto-detected), maximum trimming error rate: 0.1 (default), minimum required adapter overlap (stringency): 1 bp, minimum required sequence length for both reads before a sequence pair gets removed: 20 bp. The quality-trimmed reads were used for downstream analysis.

Raw ITS metagenomic sequencing data have been deposited in the National Center for Biotechnology Information (NCBI) Sequence Read Archive (SRA) under BioProject accession PRJNA1425464 and are publicly available at https://www.ncbi.nlm.nih.gov/sra/PRJNA1425464. The dataset includes BioSample accessions SAMN55416426-SAMN55416435.

## Isolation and morphological identification of entomopathogenic fungi

Following HTS-based assessment of fungal diversity, selective media isolation was employed to recover viable EPFs corresponding to the taxa detected. Since HTS does not yield cultivable fungi, semi-selective media containing antibiotics and fungistatic agents were used to suppress non-target microorganisms and enrich for EPFs.

Approximately 2 g of each soil sample was suspended in 1 mL sterile distilled water in Eppendorf tube, homogenized, and allowed to settle for 3–5 min. A serial dilution series was prepared by transferring 100 µL of supernatant into 900 µL sterile distilled water across five successive dilutions. From each dilution, 100 µL was spread onto Rose Bengal Chloramphenicol Agar (RBCA) plates and incubated at 25 °C in darkness for 5–10 days.

Emerging fungal colonies were initially identified macroscopically based on colony color, texture, and growth pattern [43]. Microscopic examination of conidia, phialides, and conidiophores were performed using a light microscope following established fungal identification protocols [44]. Colonies suspected to be entomopathogenic were sub-cultured onto Potato Dextrose Agar (PDA) with antibiotics, incubated at 25 °C in darkness for two weeks, and pure cultures were stored for molecular identification and bioassays.

## Pathogenicity screening and selection of fungal species

Following morphological identification, EPF isolates were selected for preliminary pathogenicity screening under laboratory conditions based on their cultural robustness, high sporulation, conidial viability, and frequent detection as abundant taxa in HTS analyses across altitudinal and land-use gradients. These criteria ensured that only isolates with strong potential for entomopathogenic activity were reserved for future detailed laboratory bioassays.

Preliminary pathogenicity screening was evaluated against second-instar larvae of *S. frugiperda*. The bioassay was designed as an initial screening to compare isolate performance under standardized laboratory conditions, rather than to estimate lethal dose parameters. Ten- to fourteen-day-old pure cultures grown on Potato Dextrose Agar (PDA) were used to prepare conidial suspensions in sterile distilled water, standardized to $1 \times 10^8$ conidia mL$^{-1}$ using a hemocytometer. Groups of 10 larvae (in triplicate) were immersed in the suspensions for 15 s and maintained under controlled laboratory conditions at 25 °C. Mortality was monitored daily for nine days, with untreated larvae serving as negative controls.

Dead larvae were surface-sterilized and incubated on moist filter paper to confirm fungal outgrowth (mycosis), thereby validating that mortality was due to fungal infection in accordance with Koch's postulates [45]. Isolates that consistently caused larval mortality and met the selection criteria were subjected to molecular confirmation and kept for future detailed bioassays.

## Genomic DNA extraction, PCR amplification and sequencing of fungal isolates

Genomic DNA was extracted from pure fungal cultures using the cetyltrimethylammonium bromide (CTAB) method with slight modifications [46]. The internal transcribed spacer (ITS) regions were amplified using universal fungal primers ITS1 (5'-TCCGTAGGTGAACCTGCGG-3') and ITS4 (5'-TCCTCCGCTTATTGATATGC-3') [47].

PCR was performed for 33 cycles with an initial denaturation at 94 °C for 30 s; denaturation at 94 °C for 30 s; annealing at 54.2 °C for 45 s; extension at 68 °C for 1 min; and a final elongation at 68 °C for 6 min. Amplicons were verified on 1% agarose gel electrophoresis run at 100 V for 90 min, stained with 0.5 µg/mL ethidium bromide, and visualized using an Alliance Uvitec gel documentation system.

Purified PCR products were submitted to Macrogen Europe B.V. (Amsterdam, Netherlands) for Sanger sequencing using an automated ABI sequencer. Raw chromatograms (.ab1) were quality-checked using Sequence Scanner Software v2.0, and low-quality 5' and 3' regions were trimmed manually. Only sequences with clear electropherograms, minimal background noise, and an average quality score (QV) ≥20 were retained. Cleaned sequences were exported in FASTA format for BLASTn-based taxonomic confirmation.

## Data collection

Morphological data included colony characteristics and microscopic features (conidia, phialides). Sequence data comprised read counts per sample, taxonomic assignments, and amplicon sequence variants. Presence/absence and relative abundance data were used to compute diversity metrics and assess distribution patterns across sites.

## Ethic statement

This study protocol was approved by the Ethics Review Board of the College of Agriculture, Sokoine University of Agriculture, Tanzania (Approval Number: SUA/DPRTC/MCS/D/2022/0021/07). Written informed consent was waived by the Ethics Review Board, as the research did not involve human participants or human data. Soil samples were collected from Sokoine University land and nearby agricultural areas with verbal permission from local authorities and landowners.

## Data analysis

All analyses were performed using R v4.4.3 [48]. Quality-trimmed metagenomic reads were taxonomically classified using the CZ ID platform (https://czid.org/), which assigns reads to taxonomic ranks from Kingdom down to Genus or Species level using its default curated reference databases (NCBI NT) [49,50]. It is important to note that CZ ID does not perform Amplicon Sequence Variants (ASV) or Operational Taxonomic Unit (OTU) inference; instead reads are classified individually based on sequence similarity. Fungal reads were extracted in R by retaining only sequences assigned to the Kingdom Fungi; while reads classified as Bacteria, Plantae, Viruses, other Eukaryota, or unclassified sequences were excluded. Reads assigned to well-documented entomopathogenic fungi (EPF) were then programmatically subset for analyses of EPF diversity, abundance, and phylogenetic relationships. To ensure accurate species-level identification, EPF reads were cross-checked against a curated reference ITS dataset compiled from GenBank. Normalized read counts exported from CZ ID were used for all downstream analyses [51].

Alpha diversity was assessed using Margalef's richness, Shannon-Wiener, Simpson's diversity, and Fisher's alpha indices. Normality was tested using the Shapiro-Wilk test, and non-normal data were log (x + 1)-transformed prior to one- or two-way ANOVA.

Beta diversity was evaluated using Bray-Curtis dissimilarity. Differences among sampling locations were tested with Permutational Multivariate Analysis of Variance (PERMANOVA) (adonis2, vegan), and homogeneity of dispersions was assessed using Permutational Analysis of Multivariate Dispersions (PERMDISP). Principal Coordinates Analysis (PCoA) based on Bray-Curtis distances was used to visualize community structure. Microbial community structure and taxa prevalence were further illustrated with $log_{10}$-transformed heatmaps and hierarchical clustering (ggplot2) to identify core and rare taxa [52].

Phylogenetic analyses were performed by aligning sequences against GenBank using BLAST. Multiple sequence alignments were conducted in MEGA v12 with the Kimura 2-parameter model, and maximum likelihood trees were constructed with 1,000 bootstrap replicates for statistical support [52–54].

## Results

### Sequence overview and taxonomic composition

Amplicon-based HTS sequencing of the ITS region yielded a total of 1,340,935 high-quality reads across the soil samples. Of these, 156,247 reads were taxonomically classified as EPF under the order Hypocreales (S4 Table). These EPF sequences were distributed among four families: Ophiocordycipitaceae (56.1%), Clavicipitaceae (37.8%), Cordycipitaceae (5.9%), and Bionectriaceae (0.1%) (Table 1).

The species composition of EPF on the Uluguru mountain slopes varied among four main families with varying species richness and occurrence frequency (Fig 2 and S5 Table). A total of 14 distinct species were detected, representing 12 genera of the four families (Table 2). Clavicipitaceae and Ophiocordycipitaceae were richer and frequently occurred with each detected in around 10 samples and comprise 4 and 5 species respectively. Bionectriaceae had moderate prevalence. Cordycipitaceae, had 4 species but with low occurrence frequency among all. The most frequently identified and abundant species included *Purpureocillium lilacinum, Metarhizium anisopliae, Pochonia chlamydosporia, Clonostachys rosea.*

### Distribution of entomopathogenic fungi species across elevation and land use

Species detection varied notably across both elevation and land use. A total of 14 EPF species were detected across the three elevations: SUA (518 m), Langali (1100 m), and Nyandira (1700 m), in both cultivated and fallow soils (Fig 3).

Species presence analysis revealed that *P. lilacinum, C. rosea*, and *M. anisopliae* were consistently detected across all altitudinal sites and nearly all cultivated soils and in fallows, *M. suchlasporia* was highly restricted to high-altitude fallow soils. *P. chlamydosporia* was detected in both fallow and cultivated soils but with varying frequency. *Lecanicillium lecanii*, *Tolypocladium*, *Beauveria felina* and *Hirsutella thompsonii* were found in low frequency and appeared only in low- or medium-altitude soils.

**Table 1. Entomopathogenic fungal families identified from HTS of Uluguru Mountain soils. Normalized read counts and relative abundance (%) are shown for each family across all sites. Values indicate proportional representation within the total ITS dataset.**

| Family | Normalized read counts | Relative abundance |
|---|---|---|
| Ophiocordicipitaceae | 87,698 | 56.10% |
| Clavicipitaceae | 59,131 | 37.80% |
| Cordycipitaceae | 9,263 | 5.90% |
| Bionectriaceae | 155 | 0.10% |
| **Total** | **156,247** | |

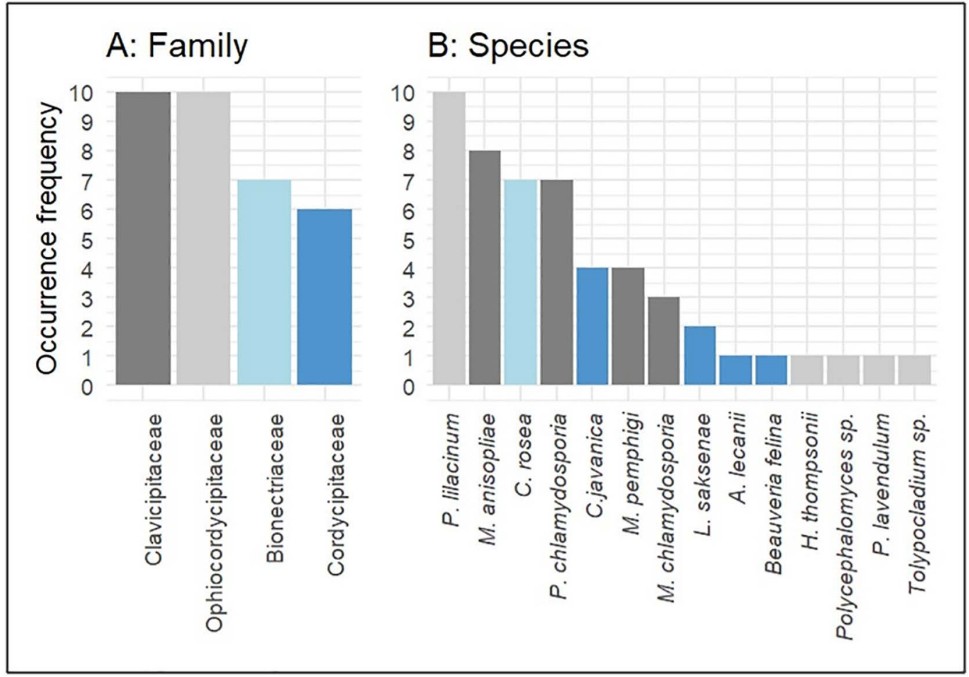

**Fig 2. Species composition and occurrence frequency of soil EPF. (A)** Occurrence frequency (%) of EPF families across all sampled sites. **(B)** Occurrence frequency of individual EPF species, with colors indicating the family to which each species belongs (as shown in **A**).

**Table 2. List of EPF species identified across sampled sites.** Species are organized by family, with sites grouped by elevation (low, medium, high). Species presence across land-use types is indicated by asterisks.

| Family | Species | Low | Medium | High |
|---|---|---|---|---|
| Ophiocordycipitaceae | *Purpureocillium lilacinum* | *** | *** | *** |
| | *Hirsutela thompsonii* | NA | * | NA |
| | *Purpureocillium lavendulum* | * | NA | NA |
| | *polycephalomyces sp.* | NA | NA | ** |
| | *Tolypocladium sp.* | ** | NA | NA |
| Clavicipitaceae | *Metarizhium anisopliae* | ** | *** | *** |
| | *Metarizhium pemphig* | NA | *** | * |
| | *Pochonia Chlamydosporia* | *** | *** | * |
| | *Metapochonia suchlasporia* | NA | NA | *** |
| Cordycipitaceae | *Cordyceps javanica* | *** | * | * |
| | *beauveria felina* | ** | NA | NA |
| | *Akanthomyces lecanii* | NA | NA | * |
| | *Lecanicillium saksenae* | NA | *** | NA |
| Bionectriaceae | *Clonostachys rosea* | *** | *** | * |

Notes: * Cultivated, ** Fallow, *** Present in both cultivated and fallow soils, NA = Not present in at all, Low = 518 m, Medium = 1100 m, High = 1700 m amsl

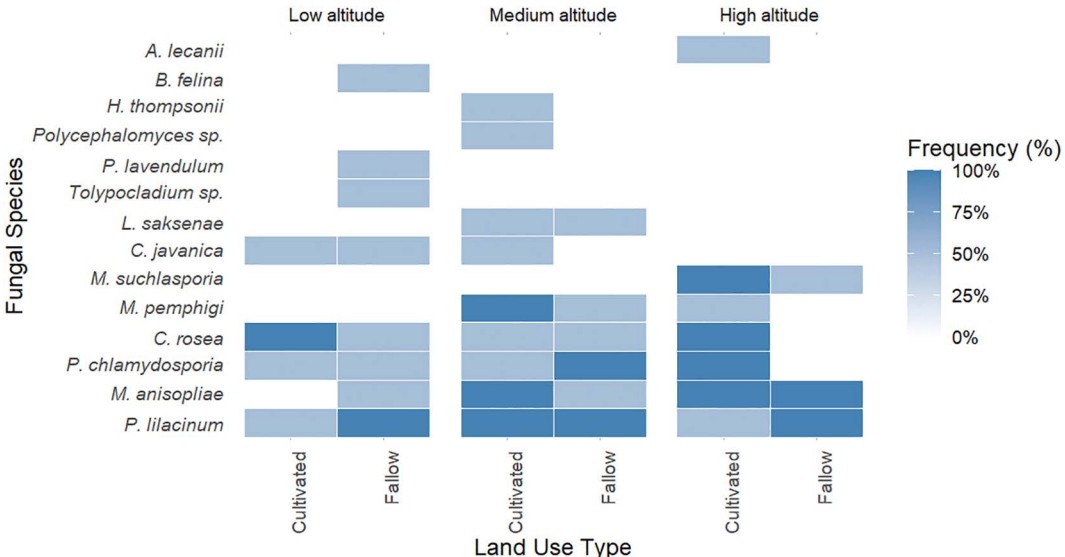

**Fig 3. Heatmap showing EPF occurrence across the altitude and land use.** Rows represent EPF species and columns represent sites grouped by elevation and land-use type (cultivated or fallow soils). Color intensity indicates relative occurrence frequency, with darker shades showing higher detection.

## Relative abundance of entomopathogenic fungal taxa

Relative abundance analysis revealed clear distribution and dominance patterns of the four families across elevation and land-use types (S1 Fig). Species dominance patterns are varied markedly across the ecological gradients (Fig 4):

At medium-altitude (Langali, 1100 m), cultivated soils were dominated by *M. anisopliae* (38.0%) and *P. lilacinum* (34.1%). High-altitude cultivated soils (Nyandira, 1700 m) harbored relative high *M. anisopliae* (30.1%) and moderate

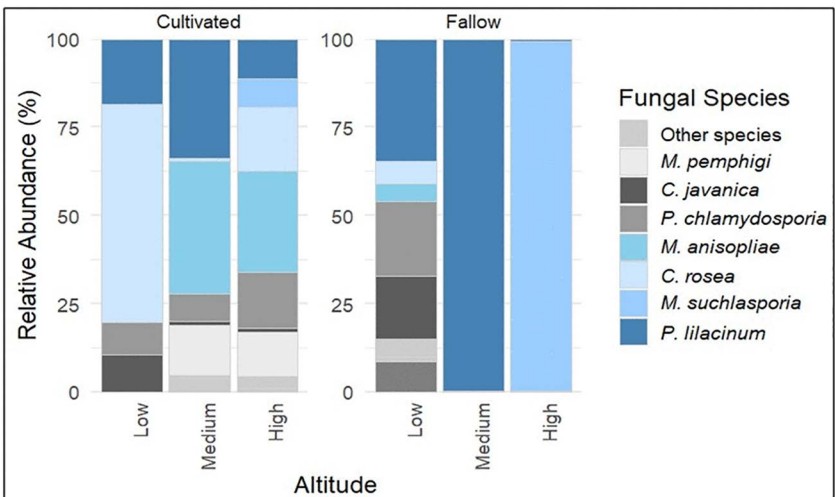

**Fig 4. Relative abundance of EPF species across the altitude and land use type.** Bar plot showing relative abundance based on normalized ITS read counts. Samples are grouped by elevation and land-use type, highlighting differences in community composition.

proportions of *C. rosea*, *P. chlamydosporia* and *M. pemphigi*. At low altitudes (SUA, 518 m), cultivated soils were mainly populated by *C. rosea* (61.9%).

In contrast, fallow soils at high altitudes were nearly monopolized by *M. suchlasporia* (99.1%), while medium-altitude fallow soils were dominated by *P. lilacinum* (99.9%) and contained a more balanced mix of *P. lilacinum* (35.2%), *P. chlamydosporia* (21.4%), and *C. javanica* (18.2%) at lower elevation.

## Species diversity in the soil across the elevation and land use

Alpha diversity indices (Shannon, Simpson, Richness, Fisher's alpha) demonstrated notable differences across both elevation and land use (Fig 5). The medium-altitude site (Langali) had the highest overall diversity, with a Shannon diversity index of 1.37 and richness of 4.75. Contrary, lowland soils (SUA) had the lowest overall diversity.

Cultivated soils generally supported higher diversity indices than fallow soils across medium and high elevations but demonstrated less variability than fallow at these sites. In contrast, lowland soils (SUA) had lower diversity in cultivated than fallow, with nearly similar variability among samples. Statistical analysis indicated that differences across altitude and land use were not significant (S1 Table), but observed trends were consistent and ecologically meaningful. To confirm that the modest sample size was sufficient to support robust statistical power, a non-parametric bootstrapping (5,000 resamples) was applied to alpha diversity indices (S6 Table). Bootstrap confidence intervals for Shannon diversity were narrow and consistent with the observed trends, confirming that the sample size provided reliable inference and that the observed patterns were not artifacts of modest sample size.

Beta Diversity and Community Clustering; PCoA based on Bray-Curtis dissimilarity showed that community composition clustered by both elevation and land use (Fig 6). A clear difference in community composition between land use types primarily revealed, accounting for 46.6% of the observed variation across the first two axes (27.5% on Axis 1 and 19.1% on Axis 2). Fallow and cultivated lands showed distinct clustering patterns, with fallow samples predominantly clustered on the positive side of Axis 1, while cultivated samples were more dispersed on the negative to zero range of this axis. Altitude, represented by different shapes, exhibited some influence but did not result in strong segregation of communities independently. Cultivated soils at medium and high-elevation showed close similarity in community structure, while low-altitude samples were more dispersed.

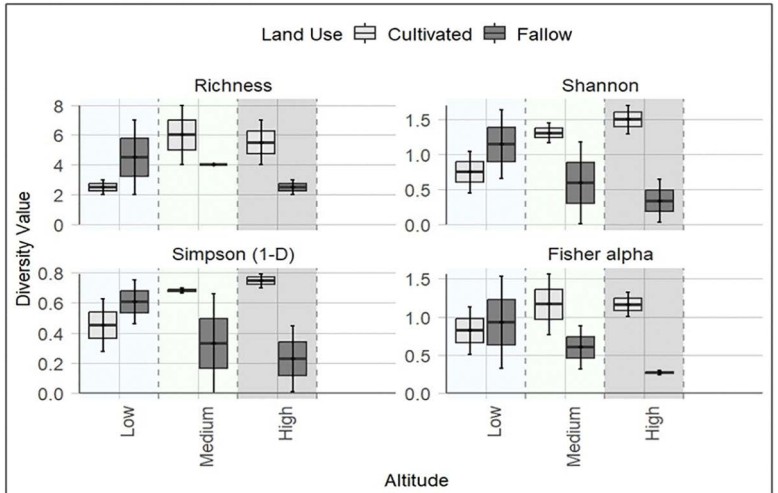

**Fig 5. The alpha diversity of EPF community across the altitude and land use.** Boxplots showing within-sample diversity using (Shannon, Simpson, richness). Boxes indicate interquartile ranges, lines show medians, and whiskers represent minima and maxima.

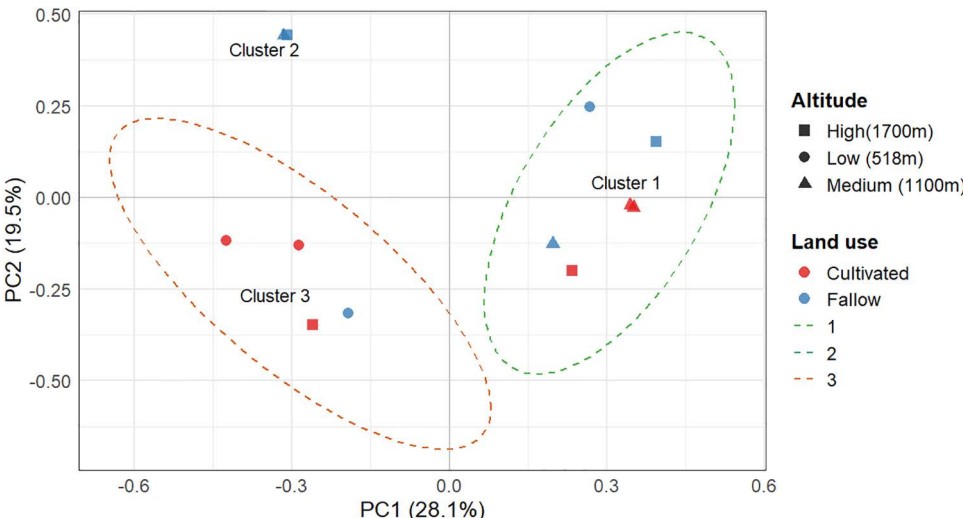

**Fig 6. Principal Coordinates Analysis (PCoA) of EPF community composition across sites differing in altitude and land use, based on Bray-Curtis dissimilarity.** Each point represents a sample, with shape indicating altitude and color indicating land-use type. PC1 (28.1%) and PC2 (19.5%) show the percentage of variation explained along each axis. Ellipses highlight k-means clusters, and cluster centroids are labeled.

Permutational multivariate analysis of variance (PERMANOVA) revealed a marginal trend in community composition across elevation and land use (pseudo-F = 2.13, R² = 0.18, df = 5, p = 0.063), indicating ecological structuring despite not reaching strict statistical significance. Homogeneity of dispersions was confirmed using PERMDISP (S2 Table). Bootstrap resampling (2,000 stratified iterations) of PERMANOVA R² values (95% CI = 0.28–0.57) and PCoA eigenvalues (Axis 1: 1.03–1.95; Axis 2: 0.58–1.25) further supported that the observed patterns were robust and ecologically meaningful, confirming that the sample size was adequate to detect meaningful community structuring.

### Selective media characterization and pathogenicity screening of fungal species

From culture-based isolation, *P. lilacinum* and *C. rosea* were the most frequently isolated genera across the study sites and easily cultured in both Rose Bengal Chloramphenicol Agar (RBCA) and PDA to produce viable and identifiable conidia. Morphologically, *P. lilacinum* colonies were lilac-colored and cottony with oval conidia, while *C. rosea* formed yellowish-white, fluffy colonies with ellipsoidal conidia (S2 Fig). A total of 12 isolates of *P. lilacinum* and 9 of *C. rosea* isolates were obtained, from which six and five representatives, respectively, were selected for preliminary pathogenicity screening against second instar of *S. frugiperda* larvae at a constant spore concentration. All tested *P. lilacinum* isolates caused infection with 15–47% mortality and 4 of 5 *C. rosea* isolates caused infection with 10–50% mortality (Table 3). This variation primarily reflects isolate-specific differences in virulence, with additional biological variability expected under laboratory bioassay conditions. Mortality data are presented as mean ± standard deviation (SD) for each isolate, providing preliminary comparative insights into isolate-specific pathogenicity. Infection symptoms included larval death and visible mycosis corresponding to species morphology. Successful re-isolation from cadaver confirmed Koch's postulates, validated their entomopathogenic potential. ITS-based phylogenetic analysis was conducted on the isolates that tested positive in the pathogenicity assays. All pathogenic isolates clustered within their respective genera, with bootstrap values ≥ 65%, and the tree was rooted using *B. bassiana* as an outgroup (see Fig 7 legend), confirming their species-level identity (Fig 7).

**Table 3. Preliminary pathogenicity of selected EPF isolates against second instar *S. frugiperda* larvae.** The table shows the number of isolates obtained per species, isolates tested, number causing positive infection, key infection signs on cadavers, mortality range (%), mean±SD, and whether re-isolation was successful. Species names are italicized.

| Species | Number of isolates | Isolates tested | Positive infection | Key infection signs | Mortality range (%) | %Mortality (Mean±SD) | Re-isolation successful |
|---|---|---|---|---|---|---|---|
| *P. lilacinum* | 12 | 6 | 6 | Larval mortality, reddish-dark cuticle mycosis | 15-47 | 31.2±12.1 | Yes |
| *C. rosea* | 9 | 5 | 4 | Dead larvae, white mycelial growth on cuticle | 20-50 | 35.0±12.9 | Yes |

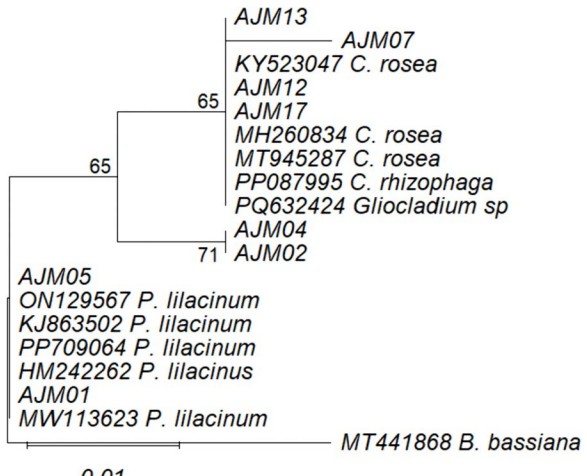

**Fig 7. Maximum likelihood phylogenetic dendrogram based on ITS rDNA sequences of soil EPF isolates.** The tree was inferred under the Kimura 2-parameter (K2) model with 1,000 bootstrap replicates. Partial deletion was applied with a 95% site coverage cutoff. Bootstrap values >= 65% are shown at the nodes. Branch lengths represent the number of substitutions per site. The tree was rooted using B. bassiana as the outgroup.

## Discussion

Our results demonstrate that soils along the Uluguru Mountain slopes harbor diverse assemblages of EPFs across multiple Hypocreales families, with Clavicipitaceae dominant and Ophiocordycipitaceae and Cordicipitaceae also represented. Both elevation and land use shaped EPF communities, confirming our hypothesis that abiotic and anthropogenic factors influence fungal diversity through soil microclimate, vegetation, organic matter, and host availability. Similar patterns have been reported in China and Pakistan [14,18,20,21,27,43,55], in Denmark [56], and in Ethiopia and Nigeria [6,34], emphasizing the role of soils particularly those enriched with litter as reservoirs for EPF persistence [19,29,57,58].

Elevation structured community composition, with medium-elevation zones showing the highest richness and turnover, forming a hump-shaped distribution. Comparable peaks at intermediate elevations have been documented in China's Changbai and Qinling Mountains [18,23,28], in other subtropical Chinese ranges [21,27], and in the Ethiopian highlands [6].

This diversity peak likely reflects moderate temperatures, balanced soil moisture, and greater habitat heterogeneity that support diverse insect hosts [22,59]. Medium-elevation zones may also act as ecological transition areas where taxa from both lower and higher altitudes overlap. These results align with ecological theory predicting maximum diversity under intermediate stress and disturbance. While elevational zonation has been widely documented in China and Europe [27,60], comparable evidence from African mountains remains scarce. Our findings provide initial evidence that Tanzanian mountain systems also exhibit hump-shaped EPFs distributions, suggesting that Medium-elevation habitats may represent areas of relatively elevated diversity within the sampled gradient; however, broader spatial replication, multi-seasonal sampling, and larger datasets are required before designating these zones as definitive biodiversity hotspots.

Land use further influenced EPFs diversity. While fallow soils are often expected to harbor greater richness [19,22,58], our results indicate that this pattern is not universal, as cultivated soils at Medium- and high-elevation sites sometimes supported higher diversity. This contrasts with studies from Pakistan and China, where uncultivated soils supported richer EPF assemblages [20,27,55]. In Ethiopia, Mekonnen et al. [6] reported more frequent recovery of *Metarhizium* and *Beauveria* in forest soils, while Hallouti et al. [61] found richer EPF communities in Moroccan argan forest soils than in citrus orchards. Local agricultural practices such as residue incorporation, fertilization, and irrigation may explain these contrasting results by enhancing soil organic matter and insect host availability [26,30,32,62]. Crops also attract insect hosts that facilitate fungal reproduction and dispersal, while certain strains remain ecologically versatile and adapted to disturbed soils [19,56]. These findings explicitly suggest that fallow soils do not always support higher EPF diversity, reflecting variability reported in previous studies. They also indicate that, cultivation does not universally suppress EPFs but its effects depend on local management and environmental conditions. This pattern is likely context-dependent and may be influenced by local agricultural practices, soil properties, and management intensity.

Across all sites, *M. anisopliae*, *P. lilacinum*, *C. rosea*, and *P. chlamydosporia* dominated, reflecting broad ecological plasticity. *Metarhizium anisopliae* is cosmopolitan, persisting across varied soils due to its wide host range, conidial resilience, and tolerance to fluctuating conditions [31,32]. *Purpureocillium lilacinum* was abundant (92% occurrence), consistent with reports from China [20,55]; its ability to colonize soils, nematode eggs, insect hosts, and rhizospheres supports persistence [62–66]. *Clonostachys rosea* thrives through multiple nutritional modes, hydrolytic enzyme production, and antimicrobial activity, while *P. chlamydosporia* exploits diverse ecological niches [67–71]. These traits underline their potential for integration into pest management programs in smallholder systems. Although species-level identification was supported by ITS-based phylogenetic analysis, the ITS marker generally offers stronger resolution at the genus level, particularly for taxa such as *Metarhizium* and *Purpureocillium*, where cryptic species complexes are common. Accordingly, future studies using multi-locus markers such as TEF1-α and RPB1/2 will be required to fully resolve species boundaries and refine taxonomic assignments.

Several taxa occurred sporadically, including *Metapochonia suchlasporia*, *Akanthomyces lecanii*, *Polycephalomyces* sp., *Tolypocladium* sp., *Beauveria felina*, *Lecanicillium saksenae*, and *Hirsutella thompsonii*. Their rarity likely reflects niche specialization or competitive exclusion by dominant taxa, consistent with findings for cryptic *Metarhizium* and rare *Lecanicillium* species in Europe [4,31,32,72]. Although infrequent, these taxa may serve as sensitive indicators of ecosystem integrity [73]. Detection of these taxa, even at low frequency, underscores the importance of conserving habitat heterogeneity and soil biodiversity.

The selective isolation and pathogenicity assays confirmed *P. lilacinum* and *C. rosea* as the most prevalent fungi, readily cultured on RBCA and PDA and morphologically consistent with earlier descriptions [71,74]. Both *P. lilacinum* and *C. rosea* showed moderate virulence against *S. frugiperda*, causing up to 47% and 50% larval mortality, respectively. These results are consistent with previous studies from China and elsewhere, which highlighted their broad entomopathogenic potential against diverse insect pests [20,64,68,69,74]. Several considerations are important when interpreting the present findings. The broad mortality range (10–50%) observed in laboratory bioassays likely reflects isolate-specific differences in virulence, together with biological heterogeneity among larvae under controlled conditions. At this stage, laboratory bioassays are best viewed as preliminary screening tools rather than definitive measures of pathogenicity and cannot directly predict field-level efficacy. In addition, the relatively limited biological replication in these initial assays may have constrained statistical resolution among isolates. This highlights the value of expanded laboratory confirmation through increased replication, larger sample sizes, and dose-response virulence assays prior to field-based evaluations, where environmental variability, host density, and soil conditions may further influence fungal performance [5,32]. Phylogenetic analysis (Fig 7) supported species-level assignments through consistent genus-level clustering (bootstrap ≥ 65%); however, the use of ITS alone may not fully resolve cryptic diversity, highlighting the importance of future multi-locus approaches. Additionally, taxonomic classification in this study relied on CZ ID rather than ASV/OTU-based inference, which may limit resolution of fine-scale diversity patterns and rare taxa detection; however, this does not affect the broader

ecological interpretations presented here. Future studies could integrate ASV/OTU-based approaches to improve resolution of fine-scale diversity and community structure.

Although the sample size was modest (n = 24), bootstrap resampling (2,000 iterations) supported the robustness of the observed ecological structuring. Nonetheless, single-season sampling and the absence of soil and host-density measurements constrain mechanistic interpretation of environmental drivers of EPF distribution. Future investigations incorporating seasonal replication, soil chemistry, and quantitative assessments of host abundance will be valuable for clarifying the ecological processes underlying observed elevational and land-use patterns. Despite these considerations, the study provides a robust baseline for understanding soil entomopathogenic fungi in Tanzanian mountain agroecosystems. Therefore, our results indicate a strong influence of both local agricultural practices and environmental conditions on EPF communities, and Uluguru Mountain slopes thus represent an important reservoir of EPFs. Harnessing this native diversity alongside improved taxonomic resolution, and laboratory and field-based validation, in future study could provide insights for biodiversity conservation and the potential for development of sustainable pest management approaches.

## Conclusion

This study demonstrates that entomopathogenic fungal communities in the Uluguru Mountains are influenced by elevation and land-use practices, with medium-elevation zones and some cultivated soils supporting the highest diversity. Dominant generalist taxa such as *M. anisopliae* and *P. lilacinum*, together with rare species, reflect both ecological resilience and conservation value. The moderate pathogenicity of *P. lilacinum* and *C. rosea* against *S. frugiperda* in laboratory assay suggests their potential for use in integrated pest management, provided their efficacy is confirmed under field conditions. These findings reinforce the importance of locally adapted EPFs as underutilized resources for sustainable agriculture and biodiversity conservation in Tanzanian mountain ecosystems. Future studies should evaluate the bio-efficacy of native isolates under field conditions, alongside seasonal dynamics, soil chemistry, and host-fungus interactions, to better integrate these fungi into agroecological pest control strategies.

## Supporting information

**S1 Table. Two-way ANOVA for diversity indices by land use, altitude and their interaction.**
(DOCX)

**S2 Table. PERMANOVA analysis across the elevations.**
(DOCX)

**S3 Table. Metadata showing sampling sites and GPS Coordinates for soil samples from different land use and altitudes.**
(DOCX)

**S4 Table. Normalized reads count assigned to specific EPF families isolated from different altitude and land use.**
(DOCX)

**S5 Table. Sequencing reads for EPF species across different samples.**
(DOCX)

**S6 Table. Non-parametric bootstraps resampling estimates of Shannon alpha diversity (BCa 95% confidence intervals) for EPF communities across altitude × land-use combinations.** Mean estimates, lower (CI low) and upper (CI high) limits are shown for each site, grouped by elevation and land-use type.
(DOCX)

**S1 Fig. Relative abundance of four Hypocreales families from different altitude and land use.**
(TIF)

**S2 Fig. Macro and micro-morphology of fungal species *C. rosea* (a-c) and *P. lilacinum* (d-f); a&d=front view b&d=reverse, c&f=Phialid structure.**
(TIF)

## Author contributions

**Conceptualization:** Abel Jonathan Mussa, Martin J Martin, Maulid W Mwatawala.

**Data curation:** Abel Jonathan Mussa.

**Formal analysis:** Abel Jonathan Mussa, Joseph O Ruboha, Sija A Kabota.

**Funding acquisition:** Maulid W Mwatawala.

**Investigation:** Abel Jonathan Mussa.

**Methodology:** Abel Jonathan Mussa, Martin J Martin.

**Resources:** Martin J Martin.

**Supervision:** Martin J Martin, Maulid W Mwatawala.

**Writing – original draft:** Abel Jonathan Mussa.

**Writing – review & editing:** Joseph O Ruboha, Sija A Kabota, Martin J Martin, Maulid W Mwatawala.

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
