## [Decision Letter · Decision Letter 0]

28 Dec 2025

PONE-D-25-54696Elevation and Land Use Shape Soil Entomopathogenic Fungal Communities in the Uluguru Mountains, TanzaniaPLOS One

Dear Dr. Mussa,

Thank you for submitting your manuscript to PLOS ONE. After careful consideration, we feel that it has merit but does not fully meet PLOS ONE’s publication criteria as it currently stands. Therefore, we invite you to submit a revised version of the manuscript that addresses the points raised during the review process.

We look forward to receiving your revised manuscript.

Kind regards,

Ebrahim Shokoohi

Academic Editor

PLOS One

Journal Requirements:

[This work was supported by the Sokoine University of Agriculture (SUA) internal research fund (Grant No. SUA/2024/AGROBIO/EPF). The funders had no role in study design, data collection and analysis, decision to publish, or preparation of the manuscript.].

Please provide an amended statement that declares *all* the funding or sources of support (whether external or internal to your organization) received during this study, as detailed online in our guide for authors at http://journals.plos.org/plosone/s/submit-now Please also include the statement “There was no additional external funding received for this study.” in your updated Funding Statement.

4. Thank you for stating the following in your manuscript:

[This work was supported by the Sokoine University of Agriculture (SUA) internal research fund (Grant No. SUA/2024/AGROBIO/EPF). The funders had no role in study design, data collection and analysis, decision to publish, or preparation of the manuscript.]

[This work was supported by the Sokoine University of Agriculture (SUA) internal research fund (Grant No. SUA/2024/AGROBIO/EPF). The funders had no role in study design, data collection and analysis, decision to publish, or preparation of the manuscript.]

5. Please upload a new copy of Figure 7 as the detail is not clear. Please follow the link for more information:  https://journals.plos.org/plosone/s/figures

6. We note that Figure 1 in your submission contains map images which may be copyrighted. All PLOS content is published under the Creative Commons Attribution License (CC BY 4.0), which means that the manuscript, images, and Supporting Information files will be freely available online, and any third party is permitted to access, download, copy, distribute, and use these materials in any way, even commercially, with proper attribution. For these reasons, we cannot publish previously copyrighted maps or satellite images created using proprietary data, such as Google software (Google Maps, Street View, and Earth). For more information, see our copyright guidelines: http://journals.plos.org/plosone/s/licenses-and-copyright.

Additional Editor Comments:

General Assessment:

The manuscript addresses an important topic—the diversity and distribution of soil-borne entomopathogenic fungi (EPFs) along elevational and land-use gradients in a biodiversity hotspot. The integration of amplicon-based metagenomics with culture-based isolation and pathogenicity assays is commendable, providing both ecological and applied perspectives. The study is well-structured, with clear hypotheses, a thorough introduction, and relevant discussion that situates the findings in a global context.

However, while the manuscript has significant merit, several issues limit its impact and reproducibility. Key concerns relate to methodological clarity, statistical robustness, phylogenetic analysis, and interpretation of results. Specific comments follow.

1. Title and Abstract:

a) The title accurately reflects the study focus. Consider including “metagenomic and culture-based approaches” to highlight methodological novelty.

b) Abstract is concise and informative, but it overstates significance in places. For example, claims about “versatile taxa offering locally adapted candidates for biological control” should be tempered, as pathogenicity data are preliminary and limited to lab assays.

2. Introduction:

a) Well-written with extensive literature coverage. The authors effectively justify the need to study EPF diversity along elevation and land-use gradients in Africa, specifically Tanzania.

b) Minor suggestion: The introduction occasionally overgeneralizes (“EPFs regulate more than 700 insect pests”) without specifying context (laboratory vs field). Consider clarifying.

c) The hypotheses are clear, but the expectation that fallow soils will always have higher diversity is not consistently supported in literature; the authors do acknowledge this later in discussion.

3. Materials and Methods:

a) Sampling Design: Sampling is clearly described. The use of 24 soil samples is modest; this limits statistical power, especially for PERMANOVA (p ≈ 0.06). Consider discussing this limitation explicitly.

b) Metagenomic Analysis:

1. ITS1-ITS2 region amplification is standard.

2. Quality control is thorough; however, the method for OTU clustering or ASV inference is not explicitly stated. Was DADA2, QIIME2, or another pipeline used? This is critical for reproducibility.

3. It is unclear how reads were assigned to EPF taxa. Were custom reference databases used for entomopathogenic fungi, or were general ITS databases applied? Misclassification of fungi is common in general ITS databases.

c) Culture-Based Isolation: Clear and methodical; semi-selective media and serial dilutions are appropriate.

d) Pathogenicity Assays:

1. Larval bioassays are well-described. However, only 10 larvae per replicate (n=3) is a small sample size for mortality estimation, reducing statistical confidence. Consider larger replicates or a more robust statistical analysis.

2. Mortality range (10–50%) is broad; authors should clarify if variability is due to isolate differences or experimental conditions.

e) Phylogenetic Analysis:

1. ITS-based phylogenies are acceptable for genus-level identification but have limited resolution at species level, especially for Metarhizium and Purpureocillium.

2. Bootstrap values or support metrics in Fig. 7 are not discussed. Many entomopathogenic fungi require multi-locus markers (TEF1, RPB1/2) to resolve cryptic species; the manuscript should acknowledge this limitation. In phylogenetic trees it must be only accession numbers and species name. All underlined, “.1” and additional signs must be removed and present the figure at standard level.

4. Results:

• Diversity Analyses:

1. Alpha diversity is reported with multiple indices. The text occasionally overinterprets non-significant trends. For example, PERMANOVA results (p = 0.063) are not statistically significant; claims about land-use shaping communities should be tempered.

2. Figures are informative, but some heatmaps and PCoA plots could benefit from clearer labeling (e.g., symbols for elevation, colors for land-use).

• Community Composition:

1. Observation of “hump-shaped” diversity at medium elevation is interesting, but limited replication and low sample size may affect robustness.

2. Relative abundance discussion is clear but could be supported with formal statistical testing (e.g., differential abundance analysis using DESeq2 or ANCOM).

• Pathogenicity and Cultivable Isolates:

1. P. lilacinum and C. rosea show moderate virulence. The discussion rightly emphasizes ecological plasticity. However, the small-scale lab assay cannot directly support field-level pest management recommendations.

5. Discussion:

a) Discussion is comprehensive, integrating results with global studies.

b) Authors appropriately highlight that cultivation may enhance EPF diversity in certain contexts, contrasting with other studies.

c) Limitations are partially addressed (small sample size, single season), but should also include:

1. ITS marker resolution limits for cryptic species identification.

2. Low number of biological replicates in pathogenicity assays.

3. Lack of direct link between soil chemistry/host density and EPF occurrence.

d) Some statements about ecological implications (e.g., “medium elevation zones are biodiversity hotspots”) are plausible but need cautious phrasing given limited data.

6. Figures and Tables:

a) Figures are generally clear; PCoA and heatmaps are informative.

b) Phylogenetic tree (Fig. 7) could be improved:

1. Include bootstrap values for all major nodes.

2. Label sequences with accession numbers to ensure reproducibility.

3. Consider adding an outgroup to properly root trees.

c) Tables are detailed. Table 3 could report standard deviation of larval mortality to better illustrate variability.

7. References:

a) Literature coverage is thorough and up-to-date.

b) Minor formatting inconsistencies observed (e.g., journal names, DOI formatting).

8. Ethical Considerations and Data Availability:

a) Ethical statement is appropriate; verbal consent for soil sampling is acceptable.

b) Public deposition of sequencing reads is commendable; consider including BioProject ID or direct SRA link in the main text.

9. Overall Strengths:

a) Integrated metagenomics and culture-based approach.

b) Comprehensive ecological interpretation.

c) Relevance for local agroecological management.

d) Clear writing and logical structure.

10. Major Concerns / Recommendations:

1. Statistical robustness: Emphasize that PERMANOVA results were non-significant; avoid overinterpreting trends.

2. Phylogenetic analysis: ITS alone may not resolve species-level relationships; clarify limitations and consider multi-locus approaches for future studies.

3. Sample size: 24 soil samples and small larval bioassays limit power; acknowledge in discussion.

4. Claims about biocontrol potential: Moderate lab-based mortality should not be interpreted as field efficacy; temper wording.

5. Methodological clarity: Provide OTU/ASV processing pipeline and reference databases.

11. Minor Suggestions:

a) Correct spelling of “Purpureocillim” → “Purpureocillium” in multiple places.

b) Improve figure labeling for clarity (colors, symbols, axis labels).

c) Include standard deviations or error bars for mortality assays.

d) Clarify whether fallow soils were recently abandoned or long-term fallows, as this affects fungal diversity.

Reviewers' comments:

Reviewer's Responses to Questions

**Comments to the Author**

1. Is the manuscript technically sound, and do the data support the conclusions?

Reviewer #1: Yes

Reviewer #2: Yes

2. Has the statistical analysis been performed appropriately and rigorously? 

Reviewer #1: Yes

Reviewer #2: No

3. Have the authors made all data underlying the findings in their manuscript fully available?

Reviewer #1: Yes

Reviewer #2: Yes

4. Is the manuscript presented in an intelligible fashion and written in standard English?

Reviewer #1: Yes

Reviewer #2: Yes

5. Review Comments to the Author

Reviewer #1: General Assessment

The manuscript presents a comprehensive investigation into how elevation and land-use gradients shape the diversity and distribution of soil-borne entomopathogenic fungi (EPFs) in the Uluguru Mountains, Tanzania. The study employs a robust combination of amplicon-based high-throughput sequencing, culture-based isolation, and pathogenicity bioassays to provide valuable ecological and applied insights. The objectives are well defined, the methodology is rigorous, and the discussion effectively links findings to previous global and regional studies.

Overall, the manuscript is of high quality and original within its regional context. While the study is scientifically rigorous and methodologically sound, I suggest that the following few minor issues and suggestions should be addressed to improve clarity, statistical interpretation, and reference consistency. In addition, I also recommend some minor editorial and formatting adjustments to enhance readability, language precision, and figure labeling.

In conclusion, this paper merits acceptance after the indicated minor revision, as it provides valuable baseline information for tropical fungal ecology, integrated pest management, and sustainable agricultural systems.

Minor Issues and Suggestions

1) The abstract is well written, though including the number of soil samples (n = 24) in the Methods section of the abstract would provide immediate clarity.

2) In the data analysis section, ensure consistent reporting of software packages with version numbers (e.g., vegan, ggplot2, MEGA v12), and cite their corresponding references.

3) In the results, the PERMANOVA finding (p = 0.06) should be described as a marginal trend and discussed in terms of ecological rather than strict statistical significance. Verify that all figures and tables (1–7) are cited in sequence and that each caption is descriptive and self-contained.

4) Minor grammatical improvements are needed to simplify some sentences and remove redundancy—for example, replacing “This study investigated” with “This study assessed.”

5) Ensure references conform to PLOS ONE style by writing full journal names and including DOIs; check for duplicate or incomplete entries, such as references 14 and 35.

6) The discussion could be further strengthened by briefly relating the observed diversity patterns to possible soil physicochemical factors such as organic matter content, pH, or moisture gradients.

Minor Editorial and Formatting Notes

1) Scientific names such as Metarhizium anisopliae and Clonostachys rosea should be italicized consistently.

2) Numerical formatting should follow scientific conventions, for example 1 × 10⁸ conidia mL⁻¹.

3) Replace approximate p-values (“p≈0.06”) with standard notation (“p = 0.06”) and ensure consistent use of hyphens and en-dashes.

Conclusion and Recommendation

In conclusion, the manuscript represents a well-designed, data-rich, and regionally important contribution to fungal ecology and biological control research. The work demonstrates that both elevation and land use significantly shape soil EPF communities and highlights promising native taxa for sustainable pest management.

Given the sound methodology, clear data presentation, and valuable ecological insight, I recommend acceptance after the above minor indicated revision.

Reviewer #2: Dear Authors

I have checked the maniscript. The manuscript addresses a relevant topic and combines ITS metabarcoding, culture-based isolation, and preliminary pathogenicity assays, which is a clear strength. The study fits the scope of PLOS ONE; however, several conclusions are stronger than supported by the data.

Key Concerns

Low sample size (n = 24) limits statistical power; most diversity metrics and PERMANOVA results are not statistically significant (p ≈ 0.06).

Despite this, the manuscript uses strong causal language (e.g., “significantly shaped”), which should be softened.

Pathogenicity assays are preliminary, and biocontrol claims should be stated cautiously.

Limitations of ITS-based species resolution, particularly for Metarhizium and related taxa, should be explicitly acknowledged.

Strengths

Addresses a clear knowledge gap in East African agroecosystems

Integrates molecular and culture-based approaches

Data availability and ethical compliance meet journal requirements

With more cautious interpretation of non-significant results and clearer acknowledgment of methodological limitations, the manuscript would be suitable for publication in PLOS ONE.

6. PLOS authors have the option to publish the peer review history of their article (what does this mean?). If published, this will include your full peer review and any attached files.

Reviewer #1: **Yes:** Lawan Adamu

Reviewer #2: No

You may also use PLOS’s free figure tool, NAAS, to help you prepare publication quality figures: https://journals.plos.org/plosone/s/figures#loc-tools-for-figure-preparation

---

## [Author Response · Author response to Decision Letter 1]

27 Feb 2026

Dear Academic Editor and Reviewers,

We sincerely thank you for your careful review and constructive feedback on our manuscript “Elevation and Land Use Shape Soil Entomopathogenic Fungal Communities in the Uluguru Mountains, Tanzania.” We greatly appreciate the time and expertise provided.

We have revised the manuscript thoroughly in response to the reviewers’ and editor’s comments. The revisions include:

Clarifying methodological details, including sequencing data processing (OTU/ASV pipeline and reference databases), sample collection, and pathogenicity assays.

Updating the Funding Statement to accurately reflect all internal funding sources and removing funding information from the Acknowledgments.

Improving statistical interpretation, clearly stating that PERMANOVA results represent marginal ecological trends due to limited sample size, and tempering claims regarding biocontrol potential.

Enhancing figure clarity, including high-resolution images, clear labeling of PCoA axes with percentage variation explained, k-means cluster ellipses, and phylogenetic trees with accession numbers and bootstrap support.

Addressing editorial and formatting issues, including spelling, consistent italicization of species names, correct numerical and p-value formatting, and updated references.

Clarifying the use of public map figures which are publicly available under CC BY 4.0 license.

We believe these revisions have strengthened the manuscript, improving clarity, reproducibility, and alignment with PLOS ONE guidelines. We hope the revised manuscript meets the journal’s standards for publication.

We sincerely thank you again for your valuable comments and consideration.

Sincerely yours,

Abel Jonathan Mussa

Corresponding author

---

## [Decision Letter · Decision Letter 1]

25 Mar 2026

PONE-D-25-54696R1Elevation and land use shape soil entomopathogenic fungal communities in the Uluguru mountains, Tanzania: Insights from metagenomic and culture-based approachesPLOS One

Dear Dr. Mussa,

Thank you for submitting your manuscript to PLOS ONE. After careful consideration, we feel that it has merit but does not fully meet PLOS ONE’s publication criteria as it currently stands. Therefore, we invite you to submit a revised version of the manuscript that addresses the points raised during the review process.

We look forward to receiving your revised manuscript.

Kind regards,

Ebrahim Shokoohi

Academic Editor

PLOS One

Journal Requirements:

Additional Editor Comments :

Dear Authors

Please check the comments by the referee, which include minor changes still needed for your manuscript.

Kind regards,

Reviewers' comments:

Reviewer's Responses to Questions

**Comments to the Author**

1. If the authors have adequately addressed your comments raised in a previous round of review and you feel that this manuscript is now acceptable for publication, you may indicate that here to bypass the “Comments to the Author” section, enter your conflict of interest statement in the “Confidential to Editor” section, and submit your "Accept" recommendation.

Reviewer #2: All comments have been addressed

2. Is the manuscript technically sound, and do the data support the conclusions?

Reviewer #2: Yes

3. Has the statistical analysis been performed appropriately and rigorously? 

Reviewer #2: Yes

4. Have the authors made all data underlying the findings in their manuscript fully available?

Reviewer #2: Yes

5. Is the manuscript presented in an intelligible fashion and written in standard English?

Reviewer #2: Yes

6. Review Comments to the Author

Reviewer #2: Dear Authors,

I would like to thank the authors for their careful and thorough revision of the manuscript. The revised version shows clear improvement in methodological clarity, statistical interpretation, and overall presentation. In particular, the authors have successfully addressed most of the major concerns raised in the previous review.

The clarification of sequencing workflow, improvement of statistical interpretation (especially regarding PERMANOVA), inclusion of data availability information, and the more cautious interpretation of pathogenicity results have significantly strengthened the manuscript.

In my opinion, the manuscript is now suitable for publication in its current form. I only have a few minor suggestions that could further improve clarity and robustness, but these can be addressed at the authors’ discretion or considered in future studies.

Minor comments and suggestions:

Bioinformatics approach

While the use of CZ ID for taxonomic classification is clearly described, it would be helpful if the authors briefly acknowledge in the discussion that the absence of ASV/OTU-based inference may influence fine-scale diversity estimates. This clarification would strengthen transparency but does not affect the overall conclusions.

Interpretation of diversity patterns

The observation that cultivated soils sometimes show higher diversity than fallow soils is interesting. A short note emphasizing that this pattern may be context-dependent (e.g., influenced by local agricultural practices or soil properties) would improve interpretation.

Pathogenicity assay

The authors have appropriately described these experiments as preliminary. For clarity, they may briefly mention in the discussion that future studies could include dose-response assays and larger sample sizes to better quantify virulence.

Minor technical corrections

Check for minor typographical issues (e.g., “OUT” vs “OTU”)

Ensure consistency in reporting statistical values and terminology

A quick check of PCR conditions (e.g., extension temperature) for clarity

7. PLOS authors have the option to publish the peer review history of their article (what does this mean?). If published, this will include your full peer review and any attached files.

Reviewer #2: No

You may also use PLOS’s free figure tool, NAAS, to help you prepare publication quality figures: https://journals.plos.org/plosone/s/figures#loc-tools-for-figure-preparation

---

## [Author Response · Author response to Decision Letter 2]

9 Apr 2026

Dear Editor,

We sincerely thank you and the reviewers for the thoughtful evaluation of our manuscript and the constructive comments provided. We have carefully considered each suggestion and revised the manuscript accordingly to improve clarity, accuracy, and readability.

A detailed, point-by-point response to each reviewer comment is included in the attached “Response to Reviewers” file. All revisions have been highlighted in the revised manuscript for transparency.

We believe that these improvements have strengthened the manuscript, and we hope it is now suitable for publication in PLOS ONE.

Thank you very much for your time and consideration.

Yours sincerely,

Abel J Mussa (Corresponding Author on behalf all Authors)

---

## [Editor Report · Decision Letter 2]

13 Apr 2026

PONE-D-25-54696R2Elevation and land use shape soil entomopathogenic fungal communities in the Uluguru mountains, Tanzania: Insights from metagenomic and culture-based approachesPLOS One

Dear Dr. Mussa,

Thank you for submitting your manuscript to PLOS ONE. After careful consideration, we feel that it has merit but does not fully meet PLOS ONE’s publication criteria as it currently stands. Therefore, we invite you to submit a revised version of the manuscript that addresses the points raised during the review process.

We look forward to receiving your revised manuscript.

Kind regards,

Ebrahim Shokoohi

Academic Editor

PLOS One

Journal Requirements:

Additional Editor Comments:

The paper improved by addressing the concerns raised by the reviewers. However, some critical point still lacking, which should be addressed. The authors must include the Bio project and SRA numbers of the sequences into the manuscript at the materials and methods, where relevant.

You may also use PLOS’s free figure tool, NAAS, to help you prepare publication quality figures: https://journals.plos.org/plosone/s/figures#loc-tools-for-figure-preparation

---

## [Author Response · Author response to Decision Letter 3]

14 Apr 2026

Dear Editor/Reviewer,

We thank you for this important comment regarding the inclusion of the BioProject and SRA numbers of the sequences. We have included the BioProject accession number (PRJNA1425464) and the corresponding BioSample accessions (SAMN55416426-SAMN55416435) in the Materials and Methods section under the high-throughput sequencing subsection. Although these accessions were already provided in the journal submission system under Data availability statement section, they have now also been incorporated into the manuscript text to ensure clarity and compliance with journal requirements. We have provided these responses in the Response to reviewers' document as well.

We believe these revisions fully address the comment and meet the journal’s requirements.

Yours sincerely,

Abel J. Mussa

(Corresponding author on behalf of all authors)

---

## [Editor Report · Decision Letter 3]

21 Apr 2026

Elevation and land use shape soil entomopathogenic fungal communities in the Uluguru mountains, Tanzania: Insights from metagenomic and culture-based approaches

PONE-D-25-54696R3

Dear Dr. Abel Jonathan Mussa,

We’re pleased to inform you that your manuscript has been judged scientifically suitable for publication and will be formally accepted for publication once it meets all outstanding technical requirements.

Kind regards,

Ebrahim Shokoohi

Academic Editor

PLOS One

Additional Editor Comments (optional):

The authors addressed all raised questions, and improved the paper.
---

## [Editor Report · Acceptance letter]

PONE-D-25-54696R3

PLOS One

Dear Dr. Mussa,

I'm pleased to inform you that your manuscript has been deemed suitable for publication in PLOS One. Congratulations! Your manuscript is now being handed over to our production team.

Kind regards,

on behalf of

Dr. Ebrahim Shokoohi

Academic Editor

PLOS One